# How Does the Family Influence Adolescent Eating Habits in Terms of Knowledge, Attitudes and Practices? A Global Systematic Review of Qualitative Studies

**DOI:** 10.3390/nu13113717

**Published:** 2021-10-22

**Authors:** Kiki S. N. Liu, Julie Y. Chen, Michelle Y. C. Ng, Maegan H. Y. Yeung, Laura E. Bedford, Cindy L. K. Lam

**Affiliations:** 1Department of Family Medicine and Primary Care, Li Ka Shing Faculty of Medicine, The University of Hong Kong, Hong Kong SAR, China; snliu@connect.hku.hk (K.S.N.L.); ngyc@connect.hku.hk (M.Y.C.N.); mhyyeung@connect.hku.hk (M.H.Y.Y.); lbedford@hku.hk (L.E.B.); clklam@hku.hk (C.L.K.L.); 2Department of Family Medicine, The University of Hong Kong Shenzhen Hospital, Shenzhen 518053, China

**Keywords:** healthy eating, adolescents, family, KAP, qualitative, review, parenting style

## Abstract

Promoting healthy eating habits can prevent adolescent obesity in which family may play a significant role. This review synthesized findings from qualitative studies to identify family barriers and facilitators of adolescent healthy eating in terms of knowledge, attitudes, and practices (KAP). A literature search of four databases was completed on 31 July 2020; qualitative studies that explored family factors of adolescent (aged 10 to 19 years) eating habits were included. A total of 48 studies were identified, with the majority being from North America and sampled from a single source. Ten themes on how family influences adolescent dietary KAP were found: Knowledge—(1) parental education, (2) parenting style, and (3) family illness experience; Attitudes—(4) family health, (5) cultivation of preference, and (6) family motivation; Practices—(7) home meals and food availability, (8) time and cost, (9) parenting style, and (10) parental practical knowledge and attitudes. This review highlights five parental characteristics underlying food parenting practices which affect adolescents’ KAP on healthy eating. Adolescents with working parents and who are living in low-income families are more vulnerable to unhealthy eating. There is a need to explore cultural-specific family influences on adolescents’ KAP, especially regarding attitudes and food choices in Asian families.

## 1. Introduction

There is strong evidence that obesity in childhood, particularly during adolescence, increases the risk of cardiovascular diseases [1,2], which are one of the most preventable causes of mortality worldwide [3]. The prevalence of adolescent obesity is rising rapidly across Asia, creating a major public health problem [4,5]. The high proportion of adolescents failing to meet dietary recommendations [6,7,8] calls for an exploration of the factors that influence the development of their eating habits.

The Knowledge, Attitudes and Practices (KAP) model is commonly applied in health education. This model emphasizes that the acquisition of knowledge is the foundation of beliefs and attitudes that reinforce the intention to adopt healthy behaviors [9]. The World Health Organization (WHO) recommends the use of the model to help identify knowledge gaps, cultural beliefs, and behavior patterns [9] in order to inform the design of effective interventions for behavior change [10]. Indeed, two studies using the KAP model found that high school students in Korea had worse dietary habits than Chinese students despite acquiring more nutritional knowledge [11,12], suggesting a translation gap from knowledge to practice. Furthermore, a recent national study of 697 Chinese adolescents (aged 12 to 17 years) showed that nutritional knowledge and social attitudes were among the major predictors for food preferences [13].

To better understand how attitudes influence eating habits, a range of theories have been applied, including the Health Belief Model, Social Cognitive Theory, and Theory of Planned Behavior [14,15,16]. These theories emphasize the role of individual perception and social influence in determining eating behavior. The most relevant concepts include: (A) *Outcome Expectation* (or *perceived benefits*) on personal health relevance [17]; (B) *Self-efficacy* (or *perceived power*) on preference towards food and preparation time [17,18]; (C) *Subjective Norms* from family and peers [18,19]; (D) *Cues to Action* from media, and the home and school environment [19]. These concepts emphasize how a person’s intention to adopt a behavior is based on cognitive and motivational beliefs, i.e., attitudes [10]. Ecological models have also been used to examine how the environment impacts eating behaviors [15], highlighting the importance of socioeconomic status, including education level and household income, community, and household food environment as external correlations of eating practices.

Indeed, the significance of food parenting practices in shaping adolescent eating habits is featured in various behavior change models. The content map by Vaughn et al. categorized food parenting practices under three constructs—coercive control, structure, and autonomy support [20]. Except for those who exert coercive control, parents who respond to adolescents’ views by supporting them structurally and autonomically will influence the social and cognitive determinants of their child’s healthy eating habits. Specifically, adolescents perceive the structure of rules and limits as well as parental modeling of dietary patterns, either healthy or not, as *Subjective Norms*. Furthermore, the structure of food availability at home, meal preparation, and snack times can promote *Cues to Action*. Autonomy support can be provided by nutrition education, child involvement, and encouragement, all of which will enhance their *Outcome Expectation* and *Self-efficacy*.

Although the influence of parental food practices has been investigated, Vaughn et al. identified a lack of research around the family, parent, and child characteristics underlying the use of these practices as well as the non-linear and indirect impact on child eating practices [20]. Indeed, certain parenting practices may be explained by latent parental characteristics, such as parenting style and knowledge [21,22,23]. Furthermore, it is important to note that parenting practices do not only affect the current eating habits of children but also influence how children choose their own food in the long term [20]. In this regard, studies should examine the impact of parent characteristics and parenting practices, collectively termed as *family factors*, on the cognition of adolescents (i.e., knowledge and attitudes that influence healthy eating practices).

Several systematic reviews of quantitative studies have explored how family factors influence eating habits among adolescents in terms of food accessibility, parental modeling, and guidance [24,25,26]. Although we know “what” family factors affect adolescent eating habits, it is not entirely clear “how” and “why” they are, or are not, able to support adolescents in adopting healthy eating practices. Qualitative studies can provide more in-depth insights into the perspectives, experiences, and attitudes that explain quantitative research findings [27]. For example, reviews by a research team in Denmark found that quantitative [28] and qualitative [29] studies complemented each other during an investigation of the determinants of fruit and vegetable intake by 6- to 18-year-olds. For example, the review of quantitative data identified socioeconomic position, parental intake, and home availability/accessibility as being positively associated with fruit and vegetable consumption (*n* = 98 studies) [28]; whereas time, cost, and several knowledge-related factors (i.e., preparation methods and short-term outcome expectations) were revealed only in the review of qualitative studies (*n* = 31) [29].

To address the research question “How does the family influence adolescent eating habits in terms of knowledge, attitudes, and practices?”, we carried out a systematic review of qualitative studies to summarize existing knowledge on family barriers and facilitators of adolescent healthy eating in terms of knowledge, attitudes, and practices, in order to identify the common family factors and any culture or social specific factors.

## 2. Materials and Methods

### 2.1. Literature Search Strategy

An electronic systematic search of the following four databases, from the year of their inception to 31 July 2020, was performed: PubMed, Web of Science, PsycINFO, and Embase. These databases covered the disciplines of life sciences, medicine, and behavioral and social sciences. References of the included papers were examined for inclusion in order to capture additional relevant articles.

The search terms were developed following the PICO framework [30] of population (adolescents and family), intervention (knowledge, attitudes, and practices), comparison (not applicable), and outcome (eating habits), which was supplemented with study design (types of qualitative data collection and analysis) (Table 1).

### 2.2. Study Selection

Articles published in English were included if they: (1) targeted healthy adolescents aged 10 to 19, determined by age range or mean age, and/or their parents/caregivers, (2) identified family factors that influence adolescent’s KAP of eating habits, and (3) applied qualitative data collection and analysis. Studies involving populations of different age groups were included only if findings specific to adolescents were described. Opinions from people other than family members, such as teachers and professionals, were included if they were relevant to family influences.

Studies were excluded if they (1) targeted adolescent groups with specific health problems (e.g., being overweight or having an eating disorder), (2) evaluated the impact of health promotion activities, such as nutrition workshops or mobile application on eating habits, (3) combined results from both quantitative and qualitative research methods, or (4) were reviews or conference abstracts.

Two reviewers independently assessed each identified article for inclusion, first based on the title and abstract, followed by an in-depth review on the full text. Discrepancies between the two reviewers were discussed to reach consensus and the first author (K.S.N.L.) further reviewed the unresolved articles.

### 2.3. Data Extraction

A data extraction excel spreadsheet was developed to systematically record all relevant data, including (1) country of origin, (2) aim(s), (3) research design, (4) sample characteristics, (5) phenomenon of interest, (6) key themes/theoretical framework, (7) major findings in KAP, and (8) conclusions. The KAP framework shown in Figure 1 was used to guide data extraction from the “Results” section of each study. The framework incorporated the key concepts of the social cognitive theories in the Attitudes construct. Parent characteristics and food parenting practices described in the content map by Vaughn et al. are the potential themes of barriers and facilitators [20]. The findings were categorized into family barriers or facilitators according to the perception of the informants. Data extraction was performed by two reviewers independently and any disagreement was resolved by a third party (K.S.N.L.).

### 2.4. Quality Assessment

The Critical Appraisal Skills Programme (CASP) checklist for qualitative studies was adopted to systematically evaluate the quality of the selected studies [31]. The checklist covered the assessment in 10 aspects, including: (1) clear statement of research aim, (2) appropriateness of qualitative research, (3) research design, (4) subject recruitment, (5) data collection methods, (6) researcher bias, (7) research ethics, (8) rigor of analysis, (9) discussion of findings, and (10) study implications. Each aspect is scored 0, 1, or 2 to denote “not complied”, “cannot tell”, and “complied”, respectively. The summation of the 10 sub-scores results in a maximum quality score of 20.

### 2.5. Data Synthesis

The findings of selected studies were first categorized according to the theoretical KAP framework shown in Figure 1. The themes on family barriers and facilitators were either deduced from the KAP framework or induced from the findings (e.g., family illness experience and cultivation of preference) by the research team. The themes were further summarized by constant comparative analysis [32], an iterative procedure to identify common or inconsistent findings across studies. Country of origin, subject characteristics (e.g., age of adolescents, type of informant), and other contextual factors of the studies were examined to identify possible reasons for inconsistent findings.

## 3. Results

### 3.1. Study Selection and Study Characteristics

A total of 2097 articles were identified from the four databases. Among the 42 articles remaining after full-text review, 6 additional articles were included following backward citation. This resulted in a final total of 48 articles from unidentical studies being included in this review. Figure 2 shows the PRISMA flow diagram on study selection. The majority of studies (*n* = 37) were published between 2010 and 2020, with the earliest article published in 2000. The studies were mainly conducted in the Americas (*n* = 31), with smaller numbers conducted in Asia (*n* = 7), Europe (*n* = 6), Africa (*n* = 2), and Australia (*n* = 1). One study was multi-national (India and Canada). Two of the Asian studies were local studies on Chinese families in Hong Kong [33,34]. Twenty-one studies interviewed multiple sources including adolescents, parents, and/or family members; fifteen of them interviewed parent-adolescent dyads from the same family; only three studies conducted the dyad interviews in the same setting [35,36,37]. Four studies targeted older adolescents aged 15 to 19 [38,39,40,41]. Study characteristics are summarized in Table 2.

### 3.2. Study Quality

The quality scores of the forty-eight reviewed studies ranged from 13 to 20 with an average of 17.8, out of a maximum of 20. Among them, 19 studies had a high quality score of 19 or above, 24 had a moderate quality score of 16 to 18, and the remaining 5 studies had a low quality score of 13 to 15 (Table 3). The studies complied with an average of 8 aspects (out of 10), ranging from 3 in one study to 10 in seven studies. All studies had a clear statement of research aim and appropriateness of qualitative research; most studies satisfied the criteria on study implications, rigor of analysis, and discussion of findings. Conversely, there was insufficient discussion on researcher bias and research design in 33 and 18 studies, respectively. The study by O’Dougherty et al. had the lowest quality score of 13 and the lowest compliance rate of only three aspects [64]; it was included in the review as the findings were consistent with other studies.

### 3.3. Synthesis of Findings

The family factors can be categorized into 10 themes under the adolescent dietary KAP framework: Knowledge—(1) nutrition education, (2) child involvement, and (3) family illness experience; Attitudes—(4) family health, (5) cultivation of preference, and (6) family motivation; Practices—(7) food preparation and availability, (8) time and cost, (9) parenting style, and (10) parental practical knowledge and attitudes. Table 4 summarizes the findings (refer to Appendix A for detailed findings extracted from each article).

#### 3.3.1. Adolescent Knowledge

There are, broadly, two types of dietary knowledge: theoretical knowledge and practical knowledge. Theoretical knowledge refers to the understanding of “why” and “what is” healthy eating, which includes health benefits and consequences of different eating habits and the recommended dietary intake. Practical knowledge refers to understanding “how” to eat healthily, from food selection during grocery shopping to cooking skills in meal preparation.


Nutrition education (theme 1)



Many studies found nutrition education by parents on both types of knowledge is a facilitator for adolescents to eat healthily, specifically, education on nutrition benefits. For example, disease prevention, weight loss and growth [38,43,51,56,66,67,70,74,77,80], health consequences of unhealthy eating such as heart and kidney problems [33,44,47,55], and healthy food choice and cooking skills [38,39,43,44,48,55,56,57,58,61,66,70,71,77,80]. However, parents with inaccurate knowledge can deliver messages that are inconsistent with what is taught at school, which confuses the adolescents [55].



Child involvement (theme 2)



Communication between parents and adolescents is important in education [37,47,54], whereas child involvement in meal planning and discussion facilitates effective communication. Four parenting strategies were identified: (1) mealtime and grocery shopping being used as opportunities to teach adolescents nutritional knowledge, such as reading food labels [38,51,55,65,66,71,80]; (2) open discussion on diet-related health outcomes [39,47]; (3) teaching adolescents about self-regulation [55,66]; (4) providing nutritional information in a casual and fun way [37]. Conversely, excluding adolescents from grocery shopping and meal preparation due to their busy study schedules served as a barrier to adolescents’ acquisition of practical nutritional knowledge [39]. Parents also perceived that the ability of food discussion was limited by their own poor eating habits [51].



Family illness experience (theme 3)



Illness experience of family members was perceived to facilitate the education of adolescents on the consequences of unhealthy eating in three US studies [44,47,55]. However, discussion on food and health was perceived as a taboo in some families [39].


#### 3.3.2. Adolescent Attitudes

Attitudes towards healthy eating were categorized into four domains: (i) outcome expectation: the belief in and the perceived importance of health outcomes resulting from eating habits; (ii) preference: the liking of certain food groups or eating habits, which includes preference for taste, fun, and health; (iii) subjective norm: expectation of significant others, such as family, peers, and teachers, by role modeling and encouragement; (iv) self-efficacy: the capability to achieve certain behavior, predominantly influenced by practical knowledge and barriers.


Family health (theme 4)



Adolescents have become more aware of the negative impacts of unhealthy eating from the experience of health problems, particularly obesity, diabetes mellitus, and heart diseases, among the family members in the US studies [37,39,47,54]. Belief in healthy eating was, on the other hand, enhanced by observing positive outcomes from family members consuming healthy foods such as fruit and vegetables [35,69].



Cultivation of preference (theme 5)



Cultivating preference for healthy food was a commonly found family facilitator, particularly in studies where participants were young adolescents [35,37,43,51,58,59,62,64,65,66,70]. Communication on nutritional information influenced their food interpretation and preference for health [37,51,59]. Parents could broaden adolescents’ taste preferences and acceptance by exposing them to a wide variety of nutritious foods at an early age [43,65,70] and by making healthy eating fun, for example, choosing vegetables in certain colors during shopping [35] as well as providing fewer food choices at home [64]. To tackle the taste preference for junk food, parents tried to reduce these temptations by limiting sedentary time and engaging their adolescents in activities such as sports and hobbies [43]. Hygienic concerns could encourage adolescents, particularly in rural areas, to eat at home instead of food stalls on the streets [60]. One barrier hindering adolescents’ preference for healthy food was family’s prioritizing either low cost or unhealthy food [40,51].



Family motivation (theme 6)



Family norm and role modeling can be both facilitators of and barriers to healthy eating in adolescents. Aside from listening to their parents’ advice, adolescents also choose their food by observing their parents’ and siblings’ eating habits of both healthy [35,37,43,44,47,55,56,57,59,66,68,70,77,80] and unhealthy foods [39,40,46,47,48,49,50,54,59,68,69,72,73,74,79]. Parents who followed the same rules as they told their children enhanced the formation of the family norm [56,80]. Parents’ verbal encouragement and compliments served as positive reinforcement for healthy eating habits [35,38,44,49,51,68,69,71,75,77]. Examples include setting expectations on diet [51,80], family support on trying healthy foods for the first time [35], making healthy eating their family lifestyle [35], describing the taste of healthy foods [66], and communication between parents regarding adolescent eating habits [80]. Parental encouragement could be initiated by their concerns around adolescents’ health, in particular weight gain and illnesses from eating unhealthy foods [57,66], which formed the subjective norm in their children [38,56,78].


In addition to family members’ preference for unhealthy foods [40,46,72], another important family barrier that impedes healthy eating was not prioritizing health in the family [42,50,51]. Some parents had cultural beliefs suggesting that thinness is a sign of sickness and convinced adolescents that body weight is not an important indicator of health [42,78], while some considered food choice was adolescents’ own responsibility, hence not providing sufficient guidance on healthy eating [42,79]. The busy schedule of working parents limited their chances to talk to their children and therefore made it difficult to encourage healthy eating among adolescents [75]. Nevertheless, two studies on low-income families found that the unhealthy eating habits in parents motivated some adolescents to eat healthily in order to be the role models for the family [77,78].

#### 3.3.3. Adolescent Practices

Practices of healthy eating refer to the consumption of fresher products such as fruit and vegetables, and less junk food with high sugar or salt content, such as sugary drinks, confectionaries, and chips. Mealtime is the major food occasion, and any food consumption between meals or after dinner is defined as snacking.


Food preparation and availability (theme 7)



Many studies identified home meals as an important facilitator of healthy eating [33,34,35,36,38,42,43,44,48,50,61,62,63,66,70,71,75,77,78,80]. Subjects regarded food prepared at home as healthy due to better variety, freshness, and reduced uses of sugar, oil, and salt [33,71,78]. Apart from home meals, food availability also influenced adolescent food choice at home. Parents could manage the food supply to provide more healthy foods and to restrict food with insufficient nutrients [36,37,39,40,42,43,44,45,47,48,49,50,52,55,56,57,59,60,61,64,65,66,67,68,69,70,73,74,77,79,80]. Failure of some families to provide healthy meals [39,54,63,76,78] or stock healthy foods at home [40,41,46,48,49,53,59,67,68,69,74,78] is explained in more depth in the following themes.



Time and cost (theme 8)



Time and cost for healthy food were important barriers to healthy food provision at home. A tight schedule and long working hours of parents prevented them from preparing family meals, encouraging adolescents to consume takeaway, fast, and prepackaged food instead of fresh foods [35,36,37,41,43,45,48,50,52,53,54,59,61,63,64,65,68,69,70,71,76,78,79,80]. Some adolescents explained that more preparation work such as washing, cutting, and cooking is required for fresh production, as compared to ready-to-eat junk food and sugary drinks [40,67,69]. Easy accessibility of restaurants and food shops further attracted families to eat unhealthy food through takeaway or when they were eating out [36,72]. Peeling fruit and vegetables as well as cutting them into ready-to-eat pieces can overcome the ‘inconvenience’ barrier, especially among young adolescents [49,56,67,69,80]. Preparing portable water in the refrigerator provided a healthier alternative to sugary drinks on hand [57,66,73], while keeping snacks in a locked cabinet restricted the accessibility to young adolescents [64].



Although low food budget might limit the purchase of junk snacks in some Western studies [36,59,66,79], it was also an important barrier to the purchase of more costly healthy food such as fruit, vegetables, and organic products, especially among low-income families who might choose unhealthy high-energy food because of its lower cost [34,35,36,37,38,41,45,46,48,49,51,54,60,61,66,67,69,72,75,77,78,79,80].Parenting style (theme 9)



Several key parenting practices were highlighted in various studies, which could be facilitating or inhibiting adolescents’ eating habits, depending on the parenting styles. Setting family rules facilitated healthy eating in adolescents during home meals and snacking. Some examples of rules include: having vegetables with every dinner, finishing everything on the plate, and serving the same meal to all family members [37,44,49,52,53,61,63,77], as well as restricting the consumption of unhealthy snacks in terms of quantity and frequency, and drinking water between glasses of juice [33,37,42,43,47,50,52,56,59,64,67,68,77,80]. Some parents further monitored their adolescents’ eating practices by verbally checking on food purchases or consumption [49,56,80], tracking food stock at home [56], and requesting adolescents to seek permission before eating unhealthy food [67]. Several authoritative parenting strategies were proposed: (i) controlling or providing supervision on food choices such as asking adolescents to at least try a few bites of healthy food [44,48,49,69,74,75]; (ii) encouraging or prompting adolescents to try healthier alternatives with reasoning [37,57,60,66,70]; (iii) having regular meal schedules and eating with family [35,52,55,71]; (iv) setting a snacking allowance [43,64]; (v) being responsive to adolescents’ preference, especially on nutritious foods [64,80]. Except for one Canadian study that found parents preferred grocery shopping alone to prevent requests for junk foods by adolescents [44], involving them in meal planning, shopping and preparation with limited/ guided choices, such as selecting from a list of food choices, and washing and cutting fresh produce [37,44,59,64,80], was perceived as a facilitator to their eating habits.Unstructured practices are a major barrier to healthy eating practices in adolescents. Accommodating taste preference of family members, usually towards fast food [37,42,64,68,78], a lack of monitoring when not eating together at a table [66], and failing to negotiate for healthy eating [79] could counteract the benefits of home meals. The lack of parental supervision also encouraged adolescents’ unhealthy snacking habits as their food choice tended to be based on taste preferences and minimal preparation effort [39,40,50,51,53,59,66,67,69,72,76,79,80]. Family members, especially grandparents, might provide adolescents with unhealthy foods as treats and bribes [36,50], and these items such as chocolates, candies, and pizza were often used as rewards for good academic performance, helping out with chores, or even eating healthy foods [53,59,66,67,72,80]. On the other hand, over-restriction could have an opposite effect on adolescents as they might want the restricted food items more and consume them on other occasions when they are not monitored [70,80]. Some adolescents with poor family relationships mentioned that the unpleasant atmosphere of eating with their family sometimes prevented them from eating at home [63].



Parental practical knowledge and attitudes (theme 10)



The perception of home meals as inferior in taste with little variation [60,62,63] could be the result of insufficient knowledge on the range of healthy food choices and inadequate skills to prepare tasty, healthy meals. To facilitate healthy eating at home, some parents highlighted the importance of cooking skills to better the presentation of healthy food, for example, hiding fruit and vegetables in soup, stew, or smoothies [43,59,66,67,80], while others would modify recipes, attempting to use healthier cooking methods, optimizing food choices, and preparing meals at lower costs [36,72]. A number of US studies reported that a high level of health awareness in parents was essential to maintaining a healthy food environment at home [40,51,58,59,63].


Parents lack of nutritional knowledge, or concern, were barriers to providing healthy meals for adolescents [34,40,50,54,71,72,77]. Disagreement on the interpretation of nutrition guidelines, such as the perceived definition of a serving among different family members [37], was an additional barrier to a healthy home food environment.

## 4. Discussion

### 4.1. Major Findings

This review of 48 qualitative studies confirmed the findings of earlier reviews on the importance of parental education, role modeling, and home food environment on adolescent eating habits [24,25,26,29], and explained how these factors influence the KAP of adolescents. In recent years (2016–2020), an increasing number of studies have explored how family facilitates healthy eating through education on health benefits, fostering cooking skills, and food discussion during mealtimes. This may be the result of more awareness of healthy eating worldwide following the launch of the 2013 Global Action Plan against non-communicable diseases by the WHO [81].

Furthermore, we found that food parenting practices described in Vaughn et al.’s content map are multifactorial and co-constructed in the family through the KAP of different members and family relationships. In particular, this review identified five parental characteristics—parental knowledge, attitudes, parenting style, time, and cost concern—as the underlying facilitators of or barriers to food parenting practices that influence adolescents’ KAP on healthy eating.

#### 4.1.1. Parental Knowledge

Parents use their knowledge to provide their children with guidance on healthy eating. Indeed, such knowledge enables parents to prepare tasty and healthy meals of varied presentation, cooking methods, and with a wide range of ingredients. These may confer further benefits by cultivating a wider taste preference and better diet quality in children, as shown in previous interventions on empowering cooking skills in parents [82,83,84].

#### 4.1.2. Parental Attitudes

Parental concern in family health facilitates motivation and provision of quality meals to adolescents. While its facilitating role is presented in this review [38,56,57,66,78], the low priority of adolescent health among parents is also revealed [42,50,78]. As proposed in a previous narrative review [85], fathers are less conscious of encouraging adolescents to eat healthily, and tend to abdicate the responsibility to their spouses, who are the primary caregivers in the family. An additional explanation is that parents are afraid of raising issues around body weight as they believe it could cause their children undue psychological stress [86]. Alternatively, some parents might perceive the increased obligations their adolescents take up from schoolwork, extracurricular activities, and social relationships with peers as more important than adolescent health. These attitudes reduce the tendency of parents to enhance dietary KAP in adolescents. Changing parents’ misconception of ‘health is less important than schoolwork’ and educating parents that a healthy diet is beneficial to school performance [87,88,89,90] can be a solution.

#### 4.1.3. Parenting Style

There is a spectrum of parenting styles, ranging from authoritarian, to authoritative, to permissive. It is a consistent finding that an authoritative parenting style (i.e., being involved with reasonable expectations) facilitates adolescents’ KAP of healthy eating—providing nutrition information in a friendly and open manner, cultivating adolescent preference for healthy eating, and controlling their food consumption by rules and monitoring—whereas the two extremes do not. Parents may encounter challenges in parenting as their children grow, especially when social impacts from peers, teachers, and media become increasingly influential, and parents are being viewed as less powerful and ideal [91]. Therefore, involving adolescents in food preparation can be a strategy to facilitate communication and exchange of knowledge and attitudes, in addition to its positive association with diet quality [92,93,94].

#### 4.1.4. Lack of Time

The competing demands from work and studies may limit the time available for families to practice healthy eating. Working mothers in Germany and the US were found to spend less time in meal preparation, eating with adolescents, and encouraging healthy eating than their non-employed or part-time counterparts [95,96]. Furthermore, the longer the length of parents’ working hours, the higher the tendency for young adolescents to have unhealthy family meals [97,98], which was reflected in this review, emphasizing that working parents preferred fast and prepackaged food instead of preparing a home meal after work.

#### 4.1.5. Cost Concern

The cost of food is another important family barrier that can impact healthy food choices. This review consistently found that many parents choose unhealthy high-energy foods for their low cost, particularly parents from low-income families. The positive relationship of socioeconomic status and diet quality is well known in the literature [99,100], where the trade-off between food cost, food waste, and health is a common dilemma [101]. One study in our review found that discussion on food choices in low SES families was understandably focused on the price of food as opposed to healthy eating and food quality [51], which may encourage adolescents to choose unhealthy food due to its lower cost. This could potentially impede their ability for behavioral change in the future.

#### 4.1.6. Effect of Study Population and Design on Results

The inclusion of studies from different cultures and informant perspectives enriched our understanding of the complex family influence on adolescent eating habits. We have identified family factors that were common to a wide spectrum of studies as well as family factors that were specific to certain cultural and demographic backgrounds.

The studies in this review had similar findings on parental roles in adolescents’ eating habits, although the discussion on family influences on adolescents’ attitudes was less comprehensive in the studies conducted in Asia. For example, whether Asian parents perceive adolescent health as important is yet to be explored. The facilitating role of family illness experiences on the knowledge and outcome expectation in adolescents was only discussed in some US studies. This may be due to the higher prevalence of obesity and non-communicable disease in the US [102,103].

Studies involving both parents and adolescents had more discussion on family facilitators, especially on adolescents’ dietary attitudes. For example, parent–adolescent dyads in two studies suggested family facilitators not seen in research involving only the parents or the adolescents, such as parents demonstrating the benefits of healthy eating [35], making healthy eating fun and a family lifestyle [35], and acquiring the habits through ongoing family discussion on healthy eating [37]. These illustrate that interviewing parent–adolescent dyads may enable a more in-depth exploration on the dyadic mutual influences in constructing eating habits.

The studies on young adolescents and their parents identified facilitators related to parenting such as family rules, parental monitoring, and parental effort in ensuring food accessibility. Conversely, studies on older adolescents mainly discussed family barriers in terms of home availability of unhealthy food and parents’ lack of time and monitoring. This may be due to the increased autonomy in adolescents and the change in the amount of parenting time. For example, a US population-based study found a decrease in parenting time from 1999 to 2004 and a transitional decrease from early to late adolescence [104]. This reflects that the time barrier becoming more common as more mothers work full-time, especially when their children grow older. Parents should be aware of how parenting time in monitoring, guiding, and providing healthy food and meals remains important in shaping adolescents’ eating habits.

### 4.2. Implications of Findings

This review demonstrates the applicability of the KAP framework in exploring familial influences on adolescents’ eating habits. It identifies the interactive effect of parental knowledge, attitudes, parenting style, and time on dietary KAP in adolescents. Some less well-recognized family factors, such as unbalanced diet due to wrong interpretation of guidelines and cultivation of food preference for health, were also explored.

Our review supports the importance of authoritative parenting and parental health concern in facilitating the development of healthy eating habits in adolescents. This counteracts the trend of adopting a permissive parenting style and a pre-mature independence of adolescents in their food choices.

Adolescents with working parents and who are living in low-income families are more vulnerable to unhealthy eating habits due to insufficient knowledge, health awareness, parenting time, and cost concern. There is a need to enhance dietary knowledge and attitudes among parents and to explore time- and cost-saving strategies of healthy eating for these vulnerable families.

### 4.3. Gaps in the Existing Knowledge

We found a disproportionately small number of studies from Asia, despite the fact that 60% of the world’s population lives in this region. Among these studies, only two targeted Chinese adolescents, both of which were from Hong Kong and interviewed only adolescents with no data from their parents. This may lead to an insufficient discussion on parental perspectives, for example, the relative importance of adolescent health among other competing concerns. This may be most relevant in Hong Kong and other Asian populations where academic achievement is greatly emphasized in the parenting culture [105] and school children are heavily burdened by academic work [106]. The relationship between how Chinese families trade off academic demands and healthy eating in their adolescents is yet to be explored.

Moreover, studies collecting perspectives from parent–adolescent dyads in the same interview are also limited to only three and none of them were on Asian populations. Dyadic interviewing enables the observation of family dynamics and how the dyads co-construct dietary KAP [107], especially parental effort in cultivating positive attitudes towards healthy eating in adolescents.

Finally, the KAP model has rarely been adopted to study family influences. Although a few studies explored adolescents’ KAP of healthy eating [37,38,61,69], none of them identified specific family influences on each of the KAP constructs. Future qualitative studies implementing the KAP model may provide additional insights on how family impacts adolescents’ knowledge of and attitudes towards healthy eating.

### 4.4. Study Limitations 

The results of this review were limited by three methodological issues. First, the studies were contextually diverse in country of origin, source of data, and SES of subjects, which could lead to bias from cultural differences despite cross-validation of findings. In order to overcome this limitation, we compared study characteristics within each theme to identify potential discrepancies between studies conducted in different contexts and minimize the synthesis bias. Second, the majority of studies did not use the KAP framework and, therefore, the results were embedded among other themes without a clear categorization using the KAP constructs. The extraction and categorization of the study findings under the KAP constructs relied on subjective interpretation by the researchers; however, we believe the use of a pre-study thematic framework and two independent reviewers can assure the validity of the results. Last, since the search was limited to studies published in English, so as to enable accessibility and peer review, some relevant studies performed on Chinese families that are published in Chinese might have been excluded.

## 5. Conclusions

Using the KAP framework, this review found that authoritative parenting styles as well as parental dietary knowledge and attitudes are facilitators of food parenting practices that promote adolescents’ KAP of healthy eating, while time and cost concerns are major barriers. Studies that conducted interviews with parent–adolescent dyads provided richer data on the dyadic mutual influence, which should be encouraged in future studies. Adolescents with working parents and low SES may be more vulnerable to unhealthy eating habits as their parents may not have sufficient knowledge and time to educate them or to serve as role models, in addition to having a limited food budget. Time- and cost-saving strategies that promote healthy eating in adolescents deserve more investigation. There could be cultural differences in family influences on adolescents’ KAP, especially in the aspects of attitudes and food choices, which call for more studies among Asian families.

## Figures and Tables

**Figure 1 nutrients-13-03717-f001:**
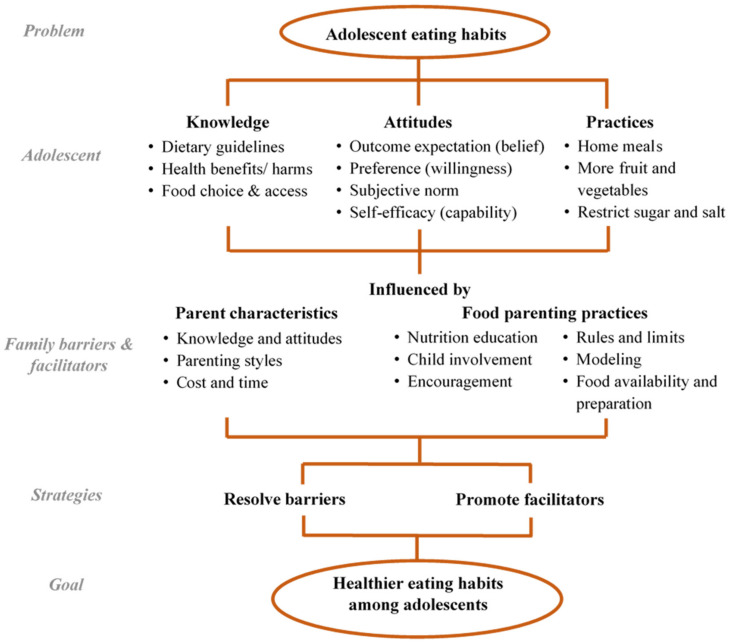
Theoretical Knowledge, Attitudes and Practices framework for data extraction on family factors.

**Figure 2 nutrients-13-03717-f002:**
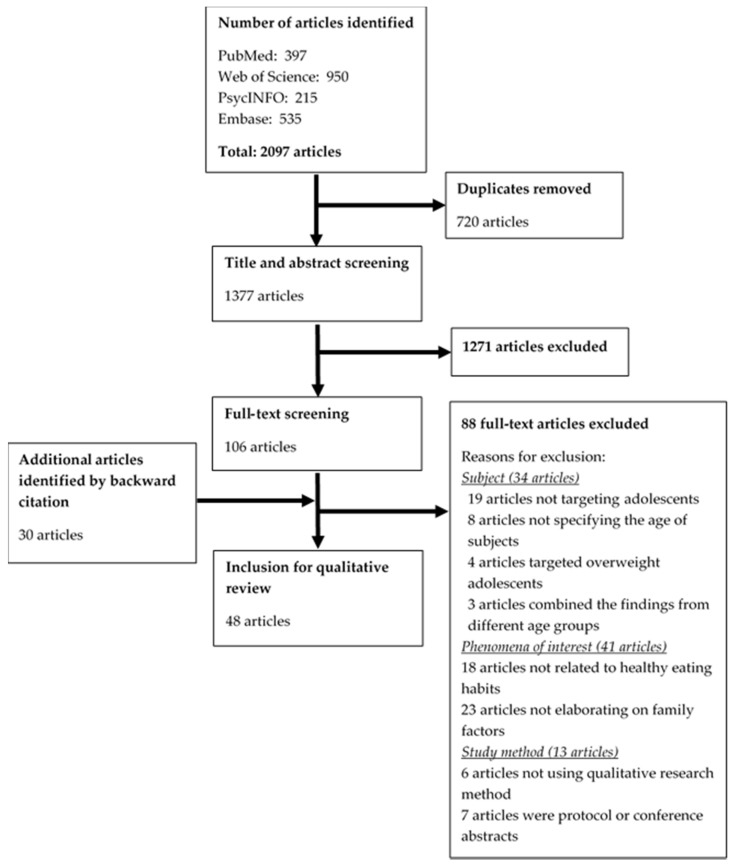
PRISMA flow diagram on study selection.

**Table 1 nutrients-13-03717-t001:** Literature search terms.

Categories	Search Terms
Adolescents and family	(adolescen * OR teen * OR youth) AND (famil * OR parent * OR home)
Knowledge, attitudes, practices	(knowledge OR literacy OR attitude * OR perception OR belie * OR willing * OR efficacy * OR ability * OR practice*)
Eating habits	(diet* OR eating habit * OR eating behavio * OR eating pattern * OR food intake OR food consumption OR food choice *)
Types of qualitative data collection and analysis	(qualitative OR interview * OR focus group *) OR (themes OR thematic OR content analys * OR framework analys * OR template analys * OR IPA OR grounded theory OR discourse analys * OR phenomenology * OR ethnograph * OR interpretative * OR inductive * OR reflexive * OR triangulat *)

Notes: The asterisk (*) represents any variation of characters.

**Table 2 nutrients-13-03717-t002:** Summary table of reviewed studies.

Study	Research Design	Sample Characteristics	Themes of the Findings on Family Factors of Adolescent KAP
Author, year, country of origin Aim(s)	Data collection method, setting, source of data, analytical method	Sample size, age and sex of adolescents (other informants indicated in brackets), sampling method	
Backett-Milburn, 2006, UK [42] To examine the perceptions and understandings underpinning the dietary practices in families with ‘normal’ weight and ‘overweight’ young teenagers living in poorer socio-economic circumstances	DI Home setting in socio-economically disadvantaged areas in Eastern Scotland Reported by parents/main food providers Inductive thematic analysis	*n* = (34) Aged 13–14 years Female = (97%) Purposive sampling by gender, BMI, and sociodemographic details	Attitudes: Family motivation (B) Practices: Food preparation and availability (F), Parenting style (F/B)
Backett-Milburn, 2010, UK [43] To understand more about the social and cultural conditions which might be promoting more positive dietary health and physical well-being amongst middle class families	DI School setting in relatively advantaged areas in Eastern Scotland Reported by parents Not specified (Inductive thematic analysis)	*n* = (35) Aged 13–14 years Female = (94%) Purposive sampling by sociodemographic details	Knowledge: Nutrition Education (F) Attitudes: Cultivation of food preference (F), Family motivation (F) Practices: Food preparation and availability (F), Time and cost (B), Parenting style (F), Parental practical knowledge and attitudes (F)
Banna et al., 2016, Peru [38] To understand socio-cultural influences on eating among adolescents in periurban Lima, Peru using qualitative methods	DI Home setting in low-income Spanish-speaking district Reported by adolescents Direct content analysis	*n* = 14 Aged 15–17 years Female = 43% Purposeful random sample from census	Knowledge: Nutrition Education (F), Child involvement (F) Attitudes: Family motivation (F) Practices: Food preparation and availability (F), Time and cost (B)
Bassett, 2008, Canada [44] To explore how adolescents and parents negotiate adolescents’ increasing food choice autonomy in European Canadian, Punjabi Canadian, and African Canadian families	DI, participant observation at a family meal and a grocery shopping trip with the family shopper(s) Home setting Reported by related adolescents and parents separately Thematic and constant comparative analyses	*n* = 47 (62) from 36 families Aged 13–19 years (mean = 41 years) Female = 72% (68%) Snowball sampling	Knowledge: Nutrition Education (F), Family illness experience (F) Attitudes: Family motivation (F) Practices: Food preparation and availability (F), Parenting style (F)
Berge et al., 2012, USA [35] To explore multiple family members’ perceptions of risk and protective factors for healthy eating and physical activity in the home.	Multi-family FG Community setting Reported by adolescents and family members, jointly Grounded hermeneutic approach	*n* = 70 (33) from 26 families Aged 8–12, 13–18 and ≥19 years (26–61 years) Female = 54% (67%) Voluntary response sampling	Attitudes: Family Health (F, Cultivation of food preference (F, Family Motivation (F) Practices: Food preparation and availability (F), Time and cost (B), Parenting style (F)
Brown et al., 2015, Botswana [45] To describe the factors that influence adolescent and adult perceptions and attitudes related to adolescent diet in Botswana	FG School setting Reported by unrelated adolescents and parents Not specified (low level, non-abstract, direct interpretation)	*n* = 12 (3) groups Aged 12–18 years Female = 6 groups (not specified) Purposive sampling by SES	Practices: Food preparation and availability (F), Time and cost (B)
Calvert et al., 2020, UK [46] To explore the perceptions of healthy eating behaviours and the influences on eating behaviours amongst 11-to-13-year-old secondary school students	Semi-structured same-sex FG Schools located in deprived areas of England Reported by adolescents Thematic framework analysis	*n* = 46 Aged 11–13 years Female = 54% Purposive sampling of selected community (low-income)	Attitudes: Family motivation (B) Practices: Food preparation and availability (B), Time and cost (B)
Chan et al., 2009, HK [33] To explore Chinese adolescents’ perceptions of healthy eating, their perceptions of various socializing agents shaping their eating habits, and their opinions about various regulatory measures which might be imposed to encourage healthy eating	FG Low to middle class population Reported by adolescents Comparison analysis method	*n* = 22 Aged 13–15 years Female = 41% Purposive sampling of selected community	Knowledge: Nutrition Education (F) Practices: Time and cost (F), Parenting style (F)
Christiansen et al., 2013, USA [47] To understand environmental factors influencing the food-related habits of low-income urban African American adolescents	DI, FG sessions, and direct observation Baltimore City Recreation Centers located in low-income predominantly African-American neighborhoods participating in Baltimore Healthy Eating Zones (BHEZ) Reported by adolescents Thematic analysis	*n* = 20 Aged 10–16 years Female = 75% Stratified, purposive sampling	Knowledge: Nutrition Education (F), Child involvement (F), Family illness experience (F) Attitudes: Family health (F), Family motivation (F/B) Practices: Food preparation and availability (F), Parenting style (F)
Correa et al., 2017, India and Canada [48] To understand perceptions and attitudes of Indian-origin adolescents in India and Canada that may contribute to healthy eating behaviour	FG School setting in rural and urban India, and urban Canada Reported by adolescents Thematic analysis with framework approach	*n* = 73 Aged 11–18 years Female = 53% Stratified, purposive sampling	Knowledge: Nutrition Education (F) Attitudes: Family motivation (B) Practices: Food preparation and availability (F/B), Time and cost (B), Parenting style (F)
Cullen, 2002, US [49] To assess social-environmental influences on children’s FJV and low-fat choices	FG by ethnicity School setting among low-income families Reported by adolescents and parents separately (relationship not specified) Not specified	*n* = 180 (40) Aged 9–12 years (mean = 36, 41, 44 years) Female = not specified (91.7–100%) Voluntary participation	Attitudes: Family motivation (F/B) Practices: Food preparation and availability (F/B), Time and cost (F/B), Parenting style (F)
Darling, 2015, US [39] To gain a deeper and more insightful understanding of the family food ecosystem, parental modeling, and parental indulgence, and their potential relationship to adolescent health and well-being	DI School setting Reported by parents Content analysis	*n* = (29) Grade 10, appx aged 15 years (35–60 years) Female: (86%) Voluntary response sampling	Knowledge: Child involvement (F/B), Family illness experience (B) Attitudes: Family health (F), Family motivation (B) Practices: Food preparation and availability (B), Time and cost (B), Parenting style (B)
Fielding-Singh, 2017, USA [50] To examine an overlooked contributor to family diet: fathers	DI Home setting in middle or upper class families Reported by related adolescents and parents separately Qualitative content analysis	*n* = 53 (56) from 44 families Aged 12–19 years Female = not specified (75%) Purposive sampling by family characteristics of interest	Attitudes: Family motivation (B) Practices: Food preparation and availability (F), Time and cost (B), Parenting style (F/B), Parental practical knowledge and attitudes (B)
Fielding-Singh and Wang, 2017, USA [51] To explore how mothers and adolescents across SES talk about food	DI Community setting across socioeconomic status Reported by related adolescents and parents separately Qualitative content analysis	*n* = 62 (62) from 62 families Aged 12–19 years Female = 61.3% (100%) Purposive and theoretical sampling by socioeconomic status	Knowledge: Nutrition Education (F), Child involvement (F/B) Attitudes: Cultivation of food preference (F/B), Family motivation (F/B) Practices: Time and cost (B), Parenting style (B). Parental practical knowledge and attitudes (F)
Fitzgerald et al., 2010, Ireland [52] To explore children’s and adolescents’ perspectives on the potential individual, social, environmental, and developmental factors that influence their food choices.	FG Primary and secondary schools Reported by adolescents Inductive thematic analysis	*n* = 29 Aged 9–18 years, mean = 13.67 Female = 55% Random sampling	Practices: Food preparation and availability (F), Time and cost (B), Parenting style (F)
Fuster et al., 2019, USA [36] To examine the perceived connections between culture and food practices among Latino pre-adolescents and their parents	DI Community setting Reported by parent-adolescent dyads/triads jointly Not specified (Content analysis)	*n* = 17 (15) from 15 families Aged 9–12 years (37.4 ± 5.1 years) Female = 47.1% (93%) Purposive sampling by ethnicity and age, and convenience sampling	Practices: Food preparation and availability (F), Time and cost (F/B), Parenting style (B), Parental practical knowledge and attitudes (F)
Garcia et al., 2019, USA [53] To increase understanding of the factors that influence Hispanic/Latino childhood obesity through an intergenerational lens including children, parents/caregivers, and grandparents.	Semi-structured FG Community based organizations in Los Angeles County Reported by adolescents Thematic analysis	*n* = 22 Aged 10–17 years Female = 52%, one unknown Not specified	Practices: Food preparation and availability (B), Time and cost (B), Parenting style (F/B)
Goh, 2009, US [54] To explore adolescent, parent, and community stakeholder perspectives on barriers to healthy eating and physical activity, and intervention ideas to address adolescent obesity	FG and DI School setting Reported by related adolescents, parents and community members separately Not specified	*n* = 119 (63) Mean age = 12 years Female = 61% (84%) Not specified	Knowledge: Child involvement (F) Attitudes: Family health (F), Family motivation (B) Practices: Food preparation and availability (B), Time and cost (B), Parental practical knowledge and attitudes (B)
Gray, 2015, Canada [55] To examine the eating behaviours and nutrition knowledge of young women in an Ontario secondary school	DI Private room within school Reported by adolescents Not specified (Thematic analysis)	*n* = 20 Aged 14–17 years Female = 100% Not specified	Knowledge: Nutrition Education (F/B), Child involvement (F), Family illness experience (F) Attitudes: Family motivation (F) Practices: Food preparation and availability (F), Parenting style (F)
Gunther et al., 2019, USA [56] To identify practices that parents use to influence early adolescents’ food choices during independent eating occasions (iEOs) from parent and child perspectives.	DI Community setting among low income families Reported by related adolescents and parents separately Directed content analysis	*n* = 44 (49) Aged 10–13 years (26–65 years) Female = 56.8% (89.8%) Purposive sampling by race and ethnicity, and convenience sampling	Knowledge: Nutrition Education (F) Attitudes: Family motivation (F) Practices: Food preparation and availability (F), Parenting style (F)
Hattersley, 2009, Australia [57] To explore adolescents’ and parents’ perceptions, attitudes, and interactions in regards to screen time (ST) and sugary drink (SD) consumption	Semi-structured FG School setting in low to middle SES areas Reported by unrelated adolescents and parents Thematic analysis	*n* = 31 (32) in 9 FG Aged 13–16 years (mean = 46 years for mothers, 43 years for fathers) Females = 42% (63%) Purposive for low-middle SES	Knowledge: Nutrition Education (F) Attitudes: Family motivation (F) Practices: Food preparation and availability (F), Parenting style (F)
Heidelberger and Smith, 2015, USA [58] To pilot Photovoice methodology with low-income, urban 9- to 13-year-olds to gain insight about their food environment and to determine whether this methodology was engaging and acceptable to them	DI Afterschool programs for youths in Supplemental Nutrition Assistance Program (SNAP)-eligible low income households Reported by adolescents Open coding method	*n* = 29 Aged 9–13 years, mean = 11 Female = 31% Not specified	Knowledge: Nutrition Education (F) Attitudes: Cultivation of food preference (F) Practices: Food preparation and availability (F), Parental practical knowledge and attitudes (F)
Holsten, 2012, US [59] To explore children’s food choices in the home with particular attention to environmental influences	DI School setting Reported by adolescents Grounded theory and content analysis	*n* = 47 Aged 11–14 years Female = 51% Maximum variation sampling by race, ethnicity, household income, and BMI	Attitudes: Cultivation of food preference (F), Family motivation (F/B) Practices: Food preparation and availability (F/B), Time and cost (F/B), Parenting style (F/B), Parental practical knowledge and attitudes (F)
Islam et al., 2019, Bangladesh [60] To explore adolescents’ and mothers’ perception of broader sociocultural aspects that sculpt the food choices, eating habits and physical activity behaviours of adolescents in Matlab, Bangladesh	FG Home setting in poor, rural areas Reported by related adolescents and parents separately Inductive thematic analysis	*n* = 4 (2) groups with 6–8 participants per group Aged 14–17 years Female = 2 groups (2) Purposive sampling	Attitudes: Cultivation of food preference (F) Practices: Food preparation and availability (F), Time and cost (B), Parenting style (F), Parental practical knowledge and attitudes (B)
Kaplan et al., 2006, USA [37] To explore how youth, parents, and grandparents discuss issues related to eating healthfully and unhealthfully and to identify intergenerational strategies for educators to improve this communication.	Semi-structured FG Community setting among low-income populations Reported by adolescents, parents and grandparents jointly Not specified (Content and ethnographic analysis)	*n* = 21 (23) from 17 families Aged 10–13 years Female = not specified (not specified) Purposive site selection and voluntary response sampling	Knowledge: Child involvement (F) Attitudes: Family health (F), Cultivation of food preference (F), Family motivation (F) Practices: Food preparation and availability (F), Time and cost (B), Parenting style (F/B), Parental practical knowledge and attitudes (B)
Kumar et al., 2016, USA [61] To enable community members to discuss their perceptions of eating habits and physical activity in relation to sixth, seventh, and eighth graders, and reveal facilitators and barriers to healthy eating behaviour and physical activity engagement.	FG Middle school among rural, limited-resources adolescents Reported by related adolescents, parents and teachers separately Not specified	*n* = 6 groups with 6–8 each (13) 6th grade to 8th grade Female = not specified (not specified) Voluntary response sampling in selected school	Knowledge: Nutrition Education (F) Practices: Food preparation and availability (F), Time and cost (B), Parenting style (F)
Monge-Rojas et al., 2005, Costa Rica [62] To assess the perceptions of rural and urban Costa Rican adolescents about their diet and the factors they consider significant to healthful eating.	FG School setting Reported by adolescents Content analysis	*n* = 108 Aged 12–18 years Female = unknown, both sex Random sampling	Attitudes: Cultivation of food preference (F) Practices: Food preparation and availability (F), Parental practical knowledge and attitudes (B)
Neumark-Sztainer et al., 2000, USA [63] To (1) increase our knowledge about family meal patterns of adolescents, (2) identify factors that adolescents perceive as reasons for not eating meals with their family, and (3) assess adolescents’ perceptions on whether they eat more healthful foods at family meals than in other eating situations.	FG School setting Reported by adolescents Content analysis	*n* = 141 Age mean = 12.6 years, 16 years Female = 61% Not specified	Practices: Food preparation and availability (F/B), Time and cost (B), Parenting style (F/B), Parental practical knowledge and attitudes (F/B)
O’dougherty et al., 2006, US [64] To gain insight into parents’ perceptions of the food preferences of their young adolescents, and their negotiating and decision-making strategies around food purchasing and meals	FG School setting among nonwhite and immigrant parents with generally low socioeconomic status Reported by parents Not specified	*n* = (52) Aged 11–15 years (mean = 37.4 years) Female: (90%) Voluntary response sampling	Attitudes: Cultivation of food preference (F) Practices: Food preparation and availability (F), Time and cost (F/B), Parenting style (F/B)
Ortega-Avila et al., 2019, Mexico [40] To explore adolescents’ perceptions of how the home environment promotes the intake of sugar-sweetened beverages (SSBs) and to identify the potential environmental cues that trigger SSB intake at home	DI Home setting or in public spaces Reported by adolescents Thematic analysis using framework approach	*n* = 29 Aged 15–19 years Female = 55% Purposive sampling by age and SSB intake	Attitudes: Cultivation of food preference (B), Family motivation (B) Practices: Food preparation and availability (B), Time and cost (B), Parenting style (B), Parental practical knowledge and attitudes (F/B)
Park et al., 2014, South Korea [65] To identify physical and social environments that influence the eating habits of adolescents living in urban South Korea	DI and FG Schools in urban areas Reported by parents and teachers Not specified (Deductive thematic analysis)	*n* = (9) Aged 10–16 years Female = (not specified) Snowball sampling	Knowledge: Child involvement (F) Attitudes: Cultivation of food preference (F) Practices: Food preparation and availability (F), Time and cost (B)
Pinard et al., 2015, USA [66] To understand influential factors related to family feeding practices among low-income English and Spanish speaking families with school-aged children when eating food away from home (FAFH)	DI Community setting Reported by related parent-adolescent separately Not specified	*n* = 20 (20) Aged 8–13 years, mean = 10.5 (<21–44 years) Female = 35% (80%) Voluntary response and snowball followed by purposive sampling by dominant language	Knowledge: Nutrition Education (F), Child involvement (F) Attitudes: Family motivation (F) Practices: Food preparation and availability (F), Time and cost (F), Parenting style (F/B)
Povey et al., 2016, UK [67] To explore the beliefs towards eating fruit and vegetables among children aged 9–11 years in a primary school setting	DI School setting in low SE group Reported by adolescents Inductive thematic analysis	*n* = 11 Aged 9–11 years, mean = 10 Females = 73% Convenience sampling	Knowledge: Nutrition Education (F) Practices: Food preparation and availability (B), Time and cost (F/B), Parenting style (F/B), Parental practical knowledge and attitudes (F)
Power et al., 2010, US [68] To provide insight into the development of a comprehensive program for the prevention of adolescent obesity: the Teen Eating and Activity Mentoring in Schools project	FG School setting from middle-class families Reported by unrelated adolescents, parents and teachers Not specified	*n* = 16 (6) Aged 12–14 years Female = 69% (67%) Not specified	Attitudes: Family motivation (F/B) Practices: Food preparation and availability (F/B), Time and cost (B), Parenting style (F/B)
Rakhshanderou et al., 2014, Iran [69] To explore the determinants of fruit and vegetable consumption among Tehranian adolescents in 2012	DI School setting Reported by adolescents Qualitative content analysis	*n* = 31 Aged 11–14 years Female = 48% Convenience sampling	Attitudes: Family health (F), Family motivation (F/B) Practices: Food preparation and availability (F/B), Time and cost (F/B), Parenting style (F/B), Parental practical knowledge and attitudes (F/B)
Rathi et al., 2016, India [70] To investigate adolescents’, parents’, teachers’, and school principals’ perceptions of the main influences on adolescent eating behaviours	DI School setting Reported by related adolescents, parents, teachers and principals separately Thematic analysis	*n* = 15 (15) Aged 14–15 years Female = 67% (93%) Purposive sampling for participating schools	Knowledge: Nutrition Education (F) Attitudes: Cultivation of food preference (F), Family motivation (F) Practices: Food preparation and availability (F), Time and cost (B), Parenting style (F/B)
Rawlins et al., 2013, UK [71] To explore both individual and family perceptions, intentions and beliefs relating to healthy lifestyles	FG and DI School and community setting Reported by related adolescents and their parents separately Thematic analysis	*n* = 70 (43) Aged 8–13 years Female = 56% (79%) Purposive sampling by ethnicity and age	Knowledge: Child involvement (F) Attitudes: Family motivation (F) Practices: Food preparation and availability (F), Time and cost (B), Parenting style (F/B), Parental practical knowledge and attitudes (B)
Rodriguez-Perez et al., 2019, Puerto Rico [72] To identify barriers that prevent healthy eating practices in Puerto Rican early adolescents (EAs).	FG Underserved community in both urban and rural settings Reported by related adolescents and parents/caregivers separately Content analysis	*n* = 52 (17) Aged 12–14 years Female = 67% (76%) Purposive sampling of sites by socioeconomic status	Attitudes: Family motivation (B) Practices: Time and cost (B), Parenting style (B), Parental practical knowledge and attitudes (F/B)
Roth-Yousey, 2012, US [73] To understand parent beverage expectations for early adolescents (EAs) by eating occasion at home and in various settings	FG School and community settings in low-income neighborhoods Reported by parents/ caregivers Not specified	*n* = (49) Aged 10–13 years Female = (86%) Purposive sampling by BMI	Attitudes: Family motivation (B) Practices: Food preparation and availability (F), Time and cost (F)
Sedibe et al., 2014, South Africa [41] To investigate the narratives pertaining to dietary and physical activity practices by female adolescents in Soweto.	Duo DI School setting Reported by adolescents Thematic analysis	*n* = 58 Aged 15.3–21.6 years (mean = 18) Female = 100% Voluntary response sampling	Practices: Food preparation and availability (B), Time and cost (B)
Sharif-Ishak et al., 2020, Malaysia [74] To explore the concepts of healthy eating and to identify the barriers and facilitating factors for dietary behaviour change in adolescents.	FG School setting Reported by adolescents Thematic analysis	*n* = 72 Aged 13–14 years Female = 48.6% Randomly selected schools followed by voluntary response sampling	Knowledge: Nutrition Education (F) Attitudes: Family motivation (B) Practices: Food preparation and availability (F/B), Parenting style (F)
Silva et al., 2015, Brazil [75] To explore how adolescents at a school in the interior of the State of Pernambuco, Brazil, perceive healthy eating	DI School setting in low HDI, agricultural region Reported by adolescents Lexical analysis	*n* = 40 Aged 10–14 years Female = 62.5% Purposive sampling	Attitudes: Family motivation (F/B) Practices: Food preparation and availability (F), Time and cost (B), Parenting style (F)
Siu et al., 2019, Hong Kong [34] To investigate the barriers to adopting healthy eating habits among secondary school students from low-income families in Hong Kong	FG School setting in a low-income district Reported by adolescents Thematic content analysis	*n* = 30 Secondary 1 and 4 Female = 50% Purposive sampling by gender, lunchbox practice, and CSSA assistance	Practices: Food preparation and availability (F), Time and cost (B), Parental practical knowledge and attitudes (B)
Snethen et al., 2007, US [76] To understand one Latino community’s perspectives about childhood overweight within this high-risk ethnic group	FG Community setting Reported by unrelated adolescents and parents Thematic analysis	*n* = 12 (24) Aged 10–12 years Female = 33% (50%) Convenience sampling	Practices: Food preparation and availability (B), Time and cost (B), Parenting style (B)
Steeves et al., 2016, US [77] To provide in-depth information on the social roles that youths’ parents and friends play related to eating and physical activity behaviours and to explore the impact of other social relationships on youths’ eating and physical activity behaviours.	DI Community setting in low-income neighbourhoods Reported by related adolescents and parents separately Direct content analysis	*n* = 38 (10) Aged 9–15 years Female = 42% (80%) Purposive sampling by genders, ages and neighbourhood locations	Knowledge: Nutrition Education (F) Attitudes: Family motivation (F) Practices: Food preparation and availability (F), Time and cost (B), Parenting style (F), Parental practical knowledge and attitudes (B)
Tiedje et al., 2014, USA [78] To describe the meanings of food, health, and wellbeing through the reported dietary preferences, beliefs, and practices of adults and adolescents from four immigrant and refugee communities in the Midwestern United States.	FG Immigrant and refugee communities Reported by adolescents Content analysis and grounded theory	*n* = 73 Aged 11–18 years Female = 53% Purposive sampling by age and gender	Attitudes: Family motivation (F/B) Practices: Food preparation and availability (F/B), Time and cost (B), Parenting style (F), Parenting style (B)
Verstraeten et al., 2014, Ecuador [79] To identify factors influencing eating behaviour of Ecuadorian adolescents—from the perspective of parents, school staff, and adolescents—to develop a conceptual framework for adolescents’ eating behaviour.	Semi-structured FG School setting in low- and middle-income countries Reported by unrelated adolescents, parents and school staff Deductive thematic content analysis	*n* = 80 (32) Aged 11–15 years (mean = 41.2 years) Female = 52.1% (75%) Convenience sampling	Attitudes: Family motivation (B) Practices: Food preparation and availability (F), Time and cost (F/B), Parenting style (B)
Zhang et al., 2018, USA [80] To explore Latino fathers’ perspectives and parenting experiences regarding early adolescents’ eating, physical activity, and screen-time behaviours using the focus group method.	FG Community setting Reported by fathers Grounded theory	*n* = (26) Aged 10–14 years (33–53 years) Female = (0%) Convenience sampling	Knowledge: Nutrition Education (F), Child involvement (F) Attitudes: Family motivation (F) Practices: Food preparation and availability (F), Time and cost (F/B), Parenting style (F/B), Parental practical knowledge and attitudes (F)

Notes: B = barrier; DI = in-depth interview; F = facilitator; FG = focus group; SES = socioeconomic status.

**Table 3 nutrients-13-03717-t003:** Quality assessment of reviewed studies (2 = Complied, 1 = Cannot tell, 0 = Not complied).

Author, Year	Q1. Was There a Clear Statement of the Aims of the Research?	Q2. Is a Qualitative Methodology Appropriate?	Q3. Was the Research Design Appropriate to Address the Aims of the Research?	Q4. Was the Recruitment Strategy Appropriate to the Aims of the Study?	Q5. Was the Data Collected in a Way that Addressed the Research Issue?	Q6. Has the Relationship between Researcher and Participants Been Adequately Considered?	Q7. Have Ethical Consideration Been Taken into Consideration?	Q8. Was the Data Analysis Sufficiently Rigorous?	Q9. Is There a Clear Statement of Findings?	Q10. How Valuable Is the Research?	Quality Score (Max = 20)
Backett-Milburn, 2006 [42]	2	2	1	2	1	1	1	2	2	2	16
Backett-Milburn, 2010 [43]	2	2	1	2	1	0	2	2	1	1	14
Banna et al., 2016 [38]	2	2	2	2	2	2	2	2	2	2	20
Bassett, 2008 [44]	2	2	1	1	1	2	2	2	2	2	17
Berge et al., 2012 [35]	2	2	2	2	2	2	2	2	2	2	20
Brown et al., 2015 [45]	2	2	1	2	2	1	2	2	2	2	18
Calvert et al., 2019 [46]	2	2	2	2	2	1	2	2	2	2	19
Chan et al., 2009 [33]	2	2	2	0	2	0	1	1	2	2	14
Christiansen et al., 2013 [47]	2	2	2	2	2	2	1	2	2	2	19
Correa et al., 2017 [48]	2	2	2	2	2	0	2	2	2	2	18
Cullen, 2002 [49]	2	2	1	1	2	1	1	1	2	2	15
Darling, 2015 [39]	2	2	2	2	2	0	2	2	2	2	18
Fielding-Singh, 2017 [50]	2	2	1	2	2	1	2	2	2	2	18
Fielding-Singh and Wang, 2017 [51]	2	2	1	2	1	1	2	2	2	2	17
Fitzgerald et al., 2010 [52]	2	2	1	1	2	1	2	2	2	2	17
Fuster et al., 2019 [36]	2	2	1	2	2	2	2	2	2	2	19
Garcia et al., 2019 [53]	2	2	1	1	2	0	2	2	2	2	16
Goh, 2009 [54]	2	2	2	1	2	1	2	2	2	2	18
Gray, 2015 [55]	2	2	2	2	1	0	2	2	2	2	17
Gunther et al., 2019 [56]	2	2	2	2	2	2	2	2	2	2	20
Hattersley, 2009 [57]	2	2	2	2	2	1	2	2	2	2	19
Heidelberger & Smith, 2015 [58]	2	2	1	0	1	1	1	2	2	2	14
Holsten, 2012 [59]	2	2	2	2	2	1	2	2	2	2	19
Islam et al., 2019 [60]	2	2	1	2	2	1	2	2	2	2	18
Kaplan et al., 2006 [37]	2	2	2	2	2	2	2	2	2	2	20
Kumar et al., 2016 [61]	2	2	2	2	2	2	2	2	2	2	20
Monge-Rojas et al., 2005 [62]	2	2	1	2	2	2	2	2	2	2	19
Neumark-Sztainer et al., 2000 [63]	2	2	2	2	1	1	1	2	2	2	17
O’Dougherty et al., 2006 [64]	2	2	1	2	1	1	1	1	1	1	13
Ortega-Avila et al., 2019 [40]	2	2	2	2	2	1	2	2	1	2	18
Park et al., 2014 [65]	2	2	1	2	2	1	2	2	2	2	18
Pinard et al., 2015 [66]	2	2	2	2	2	1	2	2	2	2	19
Povey et al., 2016 [67]	2	2	2	2	2	2	2	2	2	2	20
Power et al., 2010 [68]	2	2	2	1	1	1	2	2	2	2	17
Rakhshanderou et al., 2014 [69]	2	2	2	2	2	0	2	2	2	2	18
Rathi et al., 2016 [70]	2	2	2	2	1	2	2	2	2	2	19
Rawlins et al., 2013 [71]	2	2	2	2	2	2	2	1	2	2	19
Rodriguez-Perez et al., 2019 [72]	2	2	2	2	2	2	2	2	2	2	20
Roth-Yousey, 2012 [73]	2	2	1	1	1	2	1	2	2	2	16
Sedibe et al., 2014 [41]	2	2	2	2	2	1	2	2	2	2	19
Sharif-Ishak et al., 2020 [74]	2	2	1	1	2	1	2	2	1	2	16
Silva et al., 2015 [75]	2	2	2	1	2	1	2	2	1	2	17
Siu et al., 2018 [34]	2	2	2	2	2	0	2	2	2	2	18
Snethen et al., 2007 [76]	2	2	2	1	2	1	2	2	2	2	18
Steeves et al., 2016 [77]	2	2	2	2	2	2	1	2	2	2	19
Tiedje et al., 2014 [78]	2	2	1	2	2	1	2	2	2	2	18
Verstraeten et al., 2014 [79]	2	2	2	2	2	1	1	2	2	2	18
Zhang et al., 2018 [80]	2	2	2	2	2	1	2	2	2	2	19

**Table 4 nutrients-13-03717-t004:** Summary of findings by themes.

KAP Constructs	Family Influence		
		Facilitators	Barriers
**Adolescent Knowledge**	Nutrition education	Education on nutrition benefits [38,43,51,56,66,67,70,74,77,80] Education on undesirable consequences of unhealthy eating [33,44,47,55] Advice on healthy food choice [38,39,43,44,48,55,56,57,58,61,66,70,71,77] Fostering cooking skills [38,58,77,80]	Inconsistent message between parents and school [55]
	Child involvement	Education during mealtimes and grocery shopping [38,51,55,65,66,71,80] Good communication with adolescents (e.g., open discussion, fun style) [37,39,47,54,55,66]	Adolescents having minimal responsibility in diet-related tasks [39] Limited ability for food discussion in parents with poor role model [51]
	Family illness experience	Emphasis on diet-related health risks experienced by family members with health problems [44,47,55]	Health-related topics as a taboo in families with illnesses [39]
**Adolescent Attitudes**	Family health	Perceived importance of diet-related health risks in family members with health problems [37,39,47,54] Modeling of positive outcome of healthy eating [35,69]	
	Cultivation of preference	Cultivate taste preference for healthy food [35,37,43,51,58,59,62,64,65,70] Reducing temptation of unhealthy foods by limiting sedentary time [43] Home cooking perceived as more hygienic than street foods [60]	Family preference on low cost/ unhealthy foods [40,51]
	Family motivation	Family modeling of healthy eating habits [35,37,43,44,47,55,56,57,59,66,68,70,77,80] Encouragement and praise [35,38,44,49,51,66,68,69,71,75,77,80] Parental concern on adolescent’s health [38,56,57,66,78] Adolescent acting as a role model in the family [77,78]	Family modeling of unhealthy eating habits [39,40,46,47,48,49,50,54,59,68,69,72,73,74,79] Lack of parental concern on adolescent dietary issues (e.g., body weight is not important for the young) [42,50,51,78,79] Busy parents lacking time for encouragement [75]
**Adolescent Practices**	Food preparation and availability	Provision of healthy home meals [33,34,35,36,38,42,43,44,48,50,61,62,63,66,70,71,75,77,78,80] Home availability of healthy food/ unavailability of unhealthy foods [36,37,39,40,42,43,44,45,47,48,49,50,52,55,56,57,60,61,64,65,66,67,68,69,70,73,74,77,79,80]	Unhealthy home meals [39,54,63,76,78] Home availability of unhealthy/ unavailability of healthy foods [40,41,46,48,49,53,59,67,68,69,74,78]
	Time and cost	Controlling access (e.g., increased access to ready-to-eat FV and healthier drinks, storing unhealthy snacks out-of-reach) [49,56,57,64,66,67,69,73,80] Limiting the budget of junk food [36,59,66,79]	Convenience of packaged and takeaway foods over home cooking [35,36,37,39,43,45,48,50,52,53,54,59,61,63,64,65,68,70,76,78,79,80] and time barrier for preparing FV and healthy beverages [40,67,69] Financial preferences on affordable but unhealthy foods [34,35,36,37,38,41,45,46,48,49,51,54,60,61,66,67,69,72,75,77,78,79,80] Easy accessibility of unhealthy food shops and restaurants [36,72]
	Parenting style	Rules and monitoring of mealtimes [37,44,49,52,53,61,63,77] and snacking [33,37,42,43,47,50,52,56,59,64,67,68,77,80] Authoritative practices (e.g., reasoning, regular meal schedule) [35,37,43,44,48,49,52,55,57,59,60,64,66,69,70,71,74,75,80] Child involvement with limited/ guided choices [37,44,59,64,80] Grocery shopping without adolescents to avoid their request on unhealthy foods [44]	Unstructured practices (e.g., accommodating family taste preference, lack of monitoring at mealtimes [37,42,64,66,68,78,79] and snacking [39,40,50,51,53,59,66,67,69,76,79,80] Restriction on snacking [70,80] Treats and bribes of unhealthy foods by family members [36,50] Atmosphere of meals with dissatisfying family relation [63]
	Parental practical knowledge and attitudes	Healthy cooking skills/ varying food presentation [36,43,59,66,67,72,80] Healthy home food availability by health-conscious parents [40,51,58,59,63]	Lack of nutrition knowledge to make healthy choices or provide appropriate amounts [34,37,40,50,54,71,72,77] Lack of cooking skills to provide healthy and tasty meals with variety [60,62,63] Lack of parental concern on healthy eating [50]

Notes: FV refers to fruit and vegetables.

## Data Availability

All data generated or analyzed during this study are included in this published article and its Appendix A files.

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
