# Peer review of "How Does the Family Influence Adolescent Eating Habits in Terms of Knowledge, Attitudes and Practices? A Global Systematic Review of Qualitative Studies"

_nutrients, 2021, doi:10.3390/nu13113717_

Round 1
Reviewer 1 Report
Brief summary:
The authors reviewed the findings from qualitative studies to identify family barriers and facilitators of knowledge, attitudes and practices of healthy eating in adolescents. This review highlights five parent characteristics underlying food parenting practices which affect adolescent KAP of healthy eating.
The manuscript is well designed, written and could be interesting for the readers. Although I have some concerns about if this manuscript properly fits to the journal's scope, which I guess the editor will have taken into account and considered that the article really fits.
I have a few comments that should be addressed. There are many spelling errors that are not all covered below. Please check all spelling and grammar.
- Line 16: “literature search of four databases” Include which ones.
- Line 25: please define “SES” the first time you mentioned it.
- Please delete the doble space which are distributed in many paragraphs of the manuscript.
- Table 1: Authors should consider to divide Sample characteristics column into different columns in order to make it easier for the reader to read and understand.
Author Response
Response to Reviewer 1 Comments
Point 1: The authors reviewed the findings from qualitative studies to identify family barriers and facilitators of knowledge, attitudes and practices of healthy eating in adolescents. This review highlights five parent characteristics underlying food parenting practices which affect adolescent KAP of healthy eating.
The manuscript is well designed, written and could be interesting for the readers. Although I have some concerns about if this manuscript properly fits to the journal's scope, which I guess the editor will have taken into account and considered that the article really fits. 

Response 1: Thank you for the positive feedback.
Point 2: I have a few comments that should be addressed. There are many spelling errors that are not all covered below. Please check all spelling and grammar.
Response 2: Thank you for alerting us to this. The revised manuscript has been thoroughly proofread with any spelling or grammar errors corrected.
Point 3: Line 16: “literature search of four databases” Include which ones.
Response 3: The four databases included PubMed, Web of Science, PsycINFO and Embase. They are described under section 2.1. Literature search strategy in line 103.
Point 4: Line 25: please define “SES” the first time you mentioned it.
Response 4: Thank you for the suggestion. SES refers to socioeconomic status, which is commonly indicated by education level and household income. To be more specific we have replaced “SES” by “income” in line 24 and defined the term in its first appearance in line 57 in the main text.
Point 5: Please delete the double space which are distributed in many paragraphs of the manuscript.
Response 5: Thank you for pointing out the formatting inconsistency. We have removed the double space throughout the manuscript.
Point 6: Table 1: Authors should consider to divide Sample characteristics column into different columns in order to make it easier for the reader to read and understand.
Response 6: Thank you for the constructive suggestion, and we believe that you refer to Table 2. We understand that listing the sample characteristics by different columns may be clearer, but the increase in number of columns will reduce the space of each column for the description of the relevant details. We have added an explanation on “Sample characteristics” in the column heading of the table to clarify that they include sample size, age and sex of adolescents (other informants indicated in brackets). We hope this is acceptable.
Reviewer 2 Report
Nutrients_review October 4th, 2021
The review: “How does the family influence adolescent eating habits in terms of knowledge, attitudes and practices? A systematic review of qualitative studies” is a summary of literature from 2000-2020, which includes 48 manuscripts. The article is well-written and attempts to summarize a vast amount of information.
Major:
-the vastness of articles presents certain interpretational challenges, i.e., literature spans 20 years, where no doubt many social and economic, to name a few, influences may have occurred, as well as perspectives, changes in attitudes, please comment on the rationale here, and could this review have been just as useful using more contemporary articles, as this almost warrants it being listed in the title.
-From Table 2, included articles cover a global perspective, including: different ethnic, socioeconomic groups, mothers vs. fathers, multi-generational, multi-family vs. non, school vs. community vs. other setting, differing age groups, different geographical locations, different research designs, urban vs. rural, DI vs. FG, different reporting structures (dyads/triads), are all of these truly comparable? Even different technology is reported. This work may have been more easily reported just focusing on the U.S. manuscripts, and looking at low- vs. high income settings. By not focusing more, did this potentially dilute the findings?
-In reflection of the above-mentioned comments might authors consider a recommended title change: “How does the family influence adolescent eating habits in terms of knowledge, attitudes and practices? A global systematic review of qualitative studies (2000-2020)”
-it is clear based on the authors conclusions, that there was an attempt and objective to focus on the Asian (Chinese) populations. What is not clear is why this was not pursued from the beginning? This narrative is a bit dis-jointed for the readership, and perhaps clarification is needed by stating authors true intentions at the beginning in the introduction and should be explicitly stated.
Minor:
Line 41 should read interventional.
Line 99, from “inception”, but then later in Line 160 stated as 2000-2020. Please clarify, and perhaps maintain consistent use of language.
Table 1 search terms need to be reviewed, there are several typos and “belie” should read “belief”, “efficac” should read “efficacy”, ect…review all.
Figure 2. box on far left should read “additional” articles.
Table 2. it is not clear how the mean & median ages are reported, e.g. Bassett 2008,
“Aged 13-19 years (mean= 41 years), is this correct? Please check for all articles & throughout.
It is not clear from Table 4, who this is summated. Is it possible to quantify to some degree, how was this pulled from the articles, what it simply a mention binary Yes/No, could this be elaborated on in the methods. Perhaps include the total numbers for each out of the 48 articles, generating a value? Would be easier for readers to understand the direction of the authors.
Authors generate themes which is helpful.
Author Response
Response to Reviewer 2 Comments
Point 1: The review: “How does the family influence adolescent eating habits in terms of knowledge, attitudes and practices? A systematic review of qualitative studies” is a summary of literature from 2000-2020, which includes 48 manuscripts. The article is well-written and attempts to summarize a vast amount of information.
Response 1: Thank you for this positive feedback.
Point 2: Major:
The vastness of articles presents certain interpretational challenges, i.e., literature spans 20 years, where no doubt many social and economic, to name a few, influences may have occurred, as well as perspectives, changes in attitudes, please comment on the rationale here, and could this review have been just as useful using more contemporary articles, as this almost warrants it being listed in the title.
Response 2: Thank you for this interesting question on whether the family influence on adolescent eating habits could have changed over time. We included literature from 2000 to 2020 as we could not be sure whether and when changes might have occurred. We found no major discrepancies in the findings from studies over the years. However, we did note that, in more recent years (2016 – 2020), there has been an increasing number of studies exploring how family facilitates healthy eating through education on health benefits, fostering cooking skills and food discussion during mealtime. This may be the result of more awareness of healthy eating following the launch of the 2013 Global Action Plan against non-communicable diseases by the WHO (World Health Organization 2013). We have included a description on the changes in the literature over the years in the Discussion section (lines 382 to 386).
Reference:
World Health Organization (2013) Global Action Plan for the Prevention and Control of NCDs 2013-2020. World Health Organization, Gevena
Point 3: From Table 2, included articles cover a global perspective, including: different ethnic, socioeconomic groups, mothers vs. fathers, multi-generational, multi-family vs. non, school vs. community vs. other setting, differing age groups, different geographical locations, different research designs, urban vs. rural, DI vs. FG, different reporting structures (dyads/triads), are all of these truly comparable? Even different technology is reported. This work may have been more easily reported just focusing on the U.S. manuscripts, and looking at low- vs. high income settings. By not focusing more, did this potentially dilute the findings?
Response 3: Thank you for the constructive comments. We fully agree that the findings of qualitative studies are highly context dependent and therefore not directly comparable. While dilution of findings may be more of a concern in quantitative meta-analysis, we believe the inclusion of studies from different cultures and informant perspectives can enrich our understanding of the complex family influence on adolescent eating habits. Indeed, we have identified family factors that were common to a wide spectrum of studies as well as family factors that were specific to certain cultural and demographic backgrounds.
Point 4: In reflection of the above-mentioned comments might authors consider a recommended title change: “How does the family influence adolescent eating habits in terms of knowledge, attitudes and practices? A global systematic review of qualitative studies (2000-2020)”
Response 4: Thank you for this suggestion. We have no objection to the addition of the word “global” to the title if the editors also consider this to be appropriate.
Point 5: It is clear based on the authors conclusions, that there was an attempt and objective to focus on the Asian (Chinese) populations. What is not clear is why this was not pursued from the beginning? This narrative is a bit dis-jointed for the readership, and perhaps clarification is needed by stating authors true intentions at the beginning in the introduction and should be explicitly stated.
Response 5: Thank you for raising this issue. Our aim was to synthesize existing knowledge from a wide perspective in order to identify the common family factors that influence adolescent healthy eating. We did not intend to focus on Asian populations, although an exploration on possible culture or socioeconomic specific factors was one of our objectives. A statement regarding the study aim has been added to the Introduction (line 99). Our conclusion on “the need for more studies among Asian families” was based on the results of our review, which highlighted a disproportionately small number of studies from Asia, despite the fact that 60% of the world’s population live in this region.
Point 6: Minor:
Line 41 should read interventional.
Response 6: Thank you for this reminder. We have reworded the phrase from “effective interventional design” to “design of effective interventions”.
Point 7: Line 99, from “inception”, but then later in Line 160 stated as 2000-2020. Please clarify, and perhaps maintain consistent use of language.
Response 7: Thank you for pointing out the unclear information. The literature search was performed using the four databases from their inception and then among the 48 articles that satisfied the selection criteria, the earliest of which was published in 2000. We have reworded the relevant statement in lines 166-168 to make it clearer.
Point 8: Table 1 search terms need to be reviewed, there are several typos and “belie” should read “belief”, “efficac” should read “efficacy”, ect…review all.
Response 8: Thank you for this comment. The search terms in Table 1 refer to the terms applied to the literature search. The asterisk behind the terms denotes the inclusion of all words that start with the same letters. For example, “adolescen*” includes “adolescent”, “adolescents” and “adolescence”; while “famil*” includes “family”, “families and “familial”.
Point 9: Figure 2. box on far left should read “additional” articles.
Response 9: Thank you for pointing out this typing error. It has been corrected.
Point 10: Table 2. it is not clear how the mean & median ages are reported, e.g. Bassett 2008,
“Aged 13-19 years (mean= 41 years), is this correct? Please check for all articles & throughout.
Response 10: Thank you for pointing this out. We have added an explanation to the subheading under “Sample characteristics” in Table 2, which states that the information presented in brackets is that of informants other than the adolescents. In the study by Bassett 2008, the adolescents included in the study were aged between 13-19 years and the informants were their parents who had a mean age of 41 years.
Point 11: It is not clear from Table 4, who this is summated. Is it possible to quantify to some degree, how was this pulled from the articles, what it simply a mention binary Yes/No, could this be elaborated on in the methods. Perhaps include the total numbers for each out of the 48 articles, generating a value? Would be easier for readers to understand the direction of the authors.
Authors generate themes which is helpful.
Response 11: Thank you for the constructive comments and appreciation. Table 4 is a summary of findings synthesized from the review of the 48 studies, following the standard steps of :1) Data extraction by two reviewers independently from the “Results” section of each study guided by the KAP framework; 2) Categorization of the data into family barriers or facilitators according to the perception of the informants; 3) Data synthesis by themes deduced from the KAP framework and induced from the findings by the research team; 4) Summarization by constant comparative analysis to identify common or inconsistent findings across studies. The details are described in sections 2.3. Data extraction and 2.5. Data Synthesis. The detailed findings from each study are displayed in the Supplementary material (Table S1).
We have cited the studies relevant to each theme instead of a quantitative number of studies so that readers can refer to the details of specific studies described in Table S1. We believe this is more appropriate for the reporting on qualitative studies.
Round 2
Reviewer 2 Report
There are still a few points of clarification:
Generally speaking, please be sure to go over the final edited version before sending back to the reviewers. i.e. Lines 382-386 appear to be line 416-420? Unable to locate the edits for line 99?
Response 3: add some of this content to the discussion.
Response 4: the title needs to be more specific.
Response 5: add some of this content to the discussion.
Author Response
Thank you for the insightful review and further comments to improve our manuscript. Please find our point-by-point responses below.
Point 1: There are still a few points of clarification:
Generally speaking, please be sure to go over the final edited version before sending back to the reviewers. i.e. Lines 382-386 appear to be line 416-420? Unable to locate the edits for line 99?
Response 1: We are sorry for the unclear cross-referencing between the response to comments and the revised manuscript. The line numbers cited in the previous response referred to those in the “Simple Markup” version after removal of the tracked changes in the word file. The line numbers of the revised manuscript with tracked changes (pdf file) of the relevant revisions in Round 1 are as follows:
Lines 416-420 (response 2): “In recent years (2016 – 2020), an increasing number of studies have explored how family facilitates healthy eating through education on health benefits, fostering cooking skills and food discussion during mealtimes. This may be the result of more awareness of healthy eating worldwide following the launch of 2013 Global Action Plan against non-communicable diseases by the WHO [82].”
Lines 110-111 (response 5): “, in order to identify the common family factors and any culture or social specific factors”
Lines 180-181 (response 7): “The majority of studies (N = 37) were published between 2010 and 2020 with the earliest article published in 2000.”
Point 2: Response 3: add some of this content to the discussion.
Response 2: Thank you for the suggestion. We have added “The inclusion of studies from different cultures and informant perspectives enriched our understanding of the complex family influence on adolescent eating habits. We have identified family factors that were common to a wide spectrum of studies as well as family factors that were specific to certain cultural and demographic back-grounds.” to lines 485 to 488 under section 4.1.6. Effect of Study Population and Design on Results in the revised manuscript with tracked changes (pdf file).
Point 3: Response 4: the title needs to be more specific.
Response 3: Thank you for the suggestion. We agree to add the word “global” to indicate the inclusion of studies with a variety of contextual background, but we prefer not to include “2000-2020” since we did not limit the literature search to this period.
Point 4: Response 5: add some of this content to the discussion.
Response 4: Thank you for the suggestion. We have added “We found a disproportionately small number of studies from Asia, despite the fact that 60% of the world’s population live in this region.” to lines 533 to 534 under section 4.3 in the revised manuscript with tracked changes (pdf file).